# Promoting University Students’ Mental Health through an Online Multicomponent Intervention during the COVID-19 Pandemic

**DOI:** 10.3390/ijerph191610442

**Published:** 2022-08-22

**Authors:** Anne Theurel, Arnaud Witt, Rebecca Shankland

**Affiliations:** 1Instance Régionale D’éducation et de Promotion de la Santé, 21000 Dijon, France; 2LEAD-CNRS (UMR 5022), Université Bourgogne Franche-Comté, 21000 Dijon, France; 3Laboratory DIPHE, Department of Psychology, Education and Vulnerabilities, Université Lumière Lyon 2, 69676 Bron, France

**Keywords:** mental health, university students, self-help program, online, COVID-19

## Abstract

The mental health of university students is a serious public health issue. The alarming trend of high levels of untreated psychological distress observed during the COVID-19 pandemic highlights the need for prevention programs. Digital tools are a promising means of delivering such programs. Web-based programs are acceptable and effective at improving mental health problems and general mental well-being. However, the usefulness of such digital prevention approaches to address the multiple issues raised by the COVID-19 pandemic needs to be tested. The current study assessed the effectiveness of an 8-week online intervention, integrating a variety of evidence-based strategies for improving French university students’ mental health. Students were assigned to: (1) the online self-help program ETUCARE (*n* = 53), or (2) the control condition (*n* = 50). All the participants completed pre- and post-intervention questionnaires that assessed mental health problems and psychological well-being. The findings revealed that, compared to the control group, participation in the online program was associated with higher levels of psychological well-being post-test and fewer clinical symptoms of psychological distress, anxiety, and alcohol consumption. These preliminary findings suggest that the ETUCARE program is a promising multicomponent intervention to buffer the mental health consequences of the COVID-19 pandemic in French university students.

## 1. Introduction

The mental health of young people has long been recognized as a global public health issue [1]. Adolescence and early adulthood are considered to be the peak age for the onset of mental health disorders; three-quarters of adults with a diagnosable mental health problem will have experienced symptoms of poor mental health before the age of 25 [2]. Even before the pandemic, university students were recognized as being particularly vulnerable, with one-third experiencing significant mental health problems [3,4]. Studies indicate that university students are particularly vulnerable to stress, are at high risk for alcohol and other substance abuse [5,6,7], and experience higher levels of distress, anxiety, and depression than non-students of the same age [8]. This can be explained by demanding coursework, time pressure, poor interpersonal relationships with peers and/or lecturers [9,10,11], social isolation, peer pressure, and study–life imbalance [12]. Even in non-pandemic circumstances, students lack the necessary personal resources that could help them cope with the effects of these demands on their mental health [13]. The negative consequences of these stressors affect not only their physical and emotional health but also their academic performance, leading to increased dropout rates, poor life satisfaction, and self-confidence, and, in some cases, suicidal thoughts [14,15]. In spite of the high prevalence of mental health problems among university students, access to mental health services is often difficult, and some students may choose not to seek help even when they need mental healthcare [16,17,18]. In France, more than one in five students with a mental health problem report that they have not sought help [19], and Verger et al. [20] found that only 30.5% of students presenting one or more signs of psychological distress in the past year sought professional help.

The COVID-19 pandemic has led to major disruptions in the lives and education of university students through the prolonged closure of educational institutions, transition to internet-based learning, and social isolation from peers during state-enforced lockdowns and quarantine. Evidence shows that the pandemic has had an unprecedented impact on the mental health of university students, with increased rates of psychiatric symptoms during successive lockdowns [21,22]. Nationwide studies of university students in France carried out during successive lockdowns [23,24,25] revealed high rates of self-reported suicidal thoughts, distress, depression, anxiety and stress, and reduced access to mental healthcare, with only 12% of students with at least one symptom seeing a mental health professional and 3% using a university service [25]. Higher rates of mental health problems were reported by university students compared to age-matched controls [26]. Higher rates of alcohol consumption, eating disorders, and problematic Internet practices have also been found among stressed students [27]. COVID-19 restrictions have also contributed to poorer lifestyle behaviors [28] with a greater proportion of students experiencing physical inactivity [29], poor sleep quality [30], and unhealthy eating patterns [31,32]. Due to the sudden emergence of COVID-19, French universities cancelled all campus events and moved rapidly to transfer various courses and programmes from face-to-face to online teaching [33]. The quick closures of universities hindered the possibility to experience university life and had a major impact on academic studying. Students encountered concern about their academic future; uncertainties related to annulment/delays of activities or exams; difficulties and deficiencies related to online learning, such as the weakness of online-teaching infrastructures; the inexperience of teachers using new technologies; or the wide range of distractions when studying at home [34] Indeed, students have reported that the experience of online learning has resulted in significantly higher levels of stress and isolation as well as negative moods, and significantly lower levels of relatedness, concentration, focus, motivation, and performance compared to traditional face-to-face learning [35]. University closures also impaired the possibility to benefit from the relationships that may represent anchors in students’ lives, such as those with peers, colleagues, and professors [33,36]. The closure of campus has exacerbated academic stressors for students with increased pressures to learn independently and higher rates of anxiety and academic burn-out [37,38].

Since the onset of the COVID-19 pandemic, improving the well-being of young people and their access to mental health services has been high on the French government’s agenda. The mental health crisis prompted authorities to introduce additional measures (e.g., reimbursement of consultations with psychologists) to facilitate students’ access to mental healthcare and reduce the pressure on health systems to manageable levels [39]. However, the mental health services that are currently available are unlikely to cope with emerging demands, and such modifications represent only one building block of an organized mental health response, particularly when dealing with a pandemic such as COVID-19. To address this unprecedented mental health crisis, the public health response should involve the deployment and resourcing of: (1) secondary and tertiary prevention strategies, including treatment and preventive services for people with mental disorders, and (2) primary prevention strategies, including mental health promotion and indicated, selective, or universal prevention targeting high-risk individuals. The French High Council for Public Health (HCSP) recommended that effective evidence-based primary prevention strategies should be developed and implemented as soon as possible in order to reduce stress and psychological distress, and to avoid the increase in mental disorders and risk behaviours, such as suicidal thoughts, or addictive behaviours [40].

Prevention delivered via the internet may be a good way of engaging university students, as students today are “digital natives” who generally prefer to seek support and self-help information on the Internet [41]. Web-based mental health programs can accommodate large numbers of users, enable participants to develop skills in their own time without attending appointments, and can overcome attitudinal factors that prevent students from seeking help, including a preference for self-reliance, difficulty expressing thoughts and feelings, and stigma [18,42,43]. They are also cost-effective and scalable [44,45]. Digital mental health programs usually deliver well-established online psychological treatments that are provided as self-help material (lessons and homework assignments) so that the participants can perform most tasks independently. Some programs are entirely self-help programs without any human contact and support, while others involve a therapist’s guidance. The material is usually provided in the form of lessons or modules delivered through text, audio files, and video clips, thereby introducing participants to evidence-based therapeutic techniques and assigning them daily or weekly activities. These programs are typically comprised of 6–15 modules and usually include contents for psychoeducation and evidence-based therapeutic techniques targeting psychological processes that the developers theorize to be relevant for change. For instance, internet-based cognitive behavioural therapy (iCBT) targets maladaptive thinking patterns and aims at developing coping skills and emotional regulation. The content of iCBTs may include psychoeducation and CBT-based techniques, such as cognitive restructuring, behavioral activation, behavioral experiments, and goal setting [46]. In the same way, Internet-based acceptance and commitment therapy (iACT) primarily targets the process of psychological flexibility using ACT-based techniques, which typically focus on cognitive defusion, thereby creating acceptance instead of control, hope, and building commitment to change [47]. Before the pandemic, a number of studies have investigated different types of online mental health programs for university students [44,48,49,50] and findings suggest that these programs can improve general mental health, help prevent depression, anxiety, and alcohol-related problems [51].

Despite the call for digital mental health prevention programs to face the adverse psychological effects due to the COVID-19 pandemic [52,53], few studies tested the specific usefulness of such digital prevention approaches for university students in the context of the COVID-19 pandemic crisis [54,55]. Moreover, hardly any studies have assessed the effectiveness of a digital intervention specifically designed to target the psychological impact of the COVID-19 pandemic in the French university student population [56]. Indeed, the COVID-19 pandemic had an unprecedent impact on mental health problems and comorbidity factors, such as substance use and sleep problems. Yet, most preventative efforts have usually targeted single common problems (i.e., depression, social anxiety, substance abuse, sleep, or general well-being) [57]. Moreover, most digital interventions that have been described in studies were interventions based on one school of thought (e.g., mindfulness-based approaches) [49,57]. Hardly any studies have examined the impact of multitheoretical intervention [58]. Although recent publications have highlighted the transdiagnostic applicability of these interventions as they target shared transdiagnostic processes (e.g., maladaptive emotion regulation, experiential avoidance) that are causally interrelated with several mental disorders (i.e., depression, alcohol misuse, anxiety) [59], recent recommendations suggest the use of multitheoretical digital mental health interventions to address the multiple issues raised by the COVID-19 pandemic [60]. Multitheoretical interventions can be considered as transdiagnostic approaches combining elements from two or more evidence-based treatments into one intervention. Recent studies suggest that multitheoretical approaches may result in better outcomes than interventions from a single school of thought [60].

The present study thus aimed to investigate the effectiveness of an 8-week multitheoretical online self-help program for improving university students’ mental health during the COVID-19 pandemic. The present multitheoretical intervention involved a CBT-based focus with positive psychology and mindfulness practices and also included the paradigm of lifestyle medicine. Lifestyle medicine involves the application of lifestyle-based therapies such as positive nutrition, physical activity, sleep hygiene, stress management, and limiting or avoiding alcohol. Evidence for the effectiveness of lifestyle modifications for improving physical health is well-established [61] and recent evidence has suggested the effectiveness of lifestyle medicine for the prevention and treatment of mental health disorders [62]. A digital intervention that integrates a variety of evidence-based strategies from CBT, positive psychology, third waves approaches, and lifestyle medicine could be a good way of buffering the potential negative effects of the COVID-19 pandemic and for reducing mental health risk factors and increasing mental health promotion factors through several pathways.

The intervention of the present study is called ETUCARE, and it is an original program specifically designed to meet the specific needs of university students in the context of the COVID-19 pandemic. To this end, the ETUCARE program was co-designed with university students using a participatory approach based on an online survey and on Zoom meetings with students with the aim to include content and features that were relevant to them [63]. We hypothesized that university students who completed the ETUCARE program would have less mental health problems, such as psychological distress, anxiety and depressive symptoms, less alcohol use and sleep problems, and higher levels of psychological well-being post-test, compared to those in the control group.

## 2. Methods

### 2.1. Study Design

A two-armed, non-randomized controlled trial was conducted to determine the efficacy of an online self-help program on the mental health and well-being of university students. The intervention group was composed of participants who received the ETUCARE program, and the control group comprised students on a waiting list who did not follow the program during the study period.

### 2.2. Participants and Procedure

The inclusion criteria were enrolment at the university, to be fluent in French, being aged between 18 and 30 years old, and to have an e-mail address and access to the Internet at home or through a smartphone. Recruitment was based on voluntary participation and was made through university health and social services and class announcements. No monetary compensation or other incentive was offered to participants. All participants signed an informed consent and were told that their participation was voluntary and could be withdrawn at any time, and that their information would remain anonymous. After signing the consent form, participants received a secret code to ensure confidentiality of the collected data. All data were collected online and were stored on a secure computer. A total of 107 students expressed interest in participating in the study. The pre-test (T0) was completed by 53 participants in the intervention group and 50 in the control group (*N* = 103). Most participants were students in psychology (*n* = 55) and others came from various disciplines (6 in law, 4 in literature and languages, 2 in philosophy, 3 in history, 2 in physical activity and sports, 2 in arts, 10 in biology, 16 in medicine, and 3 in agri-food). The initial sample was composed of 18 men (10 in the intervention group and 8 in the control group) and 85 women (43 in the intervention group and 42 in the control group), with a mean age of 20.19 years for participants in the intervention group and 19.72 for participants in the control group.

After dropout, our final sample was composed of 58 students, which represents 54% of the initial sample. The intervention group contained participants who completed post-test assessments and were assigned to the ETUCARE program (*n* = 20; 80% female; Mage = 20.3; Age range: 18–25 years) and the control group was composed of participants who completed post-test assessments and did not take part in the program (*n* = 38; 85.2% female; Mage = 19.8; Age range: 18–23 years). In the intervention group, students who dropped out did not differ with students with complete cases after the intervention on age and on mental health measures (all F’s < 1). In the control group, students who dropped out had higher levels of psychological distress, *F* (1, 48) = 4.44, *p* < 0.05, and presented more depressive symptoms, *F* (1, 48) = 5.19, *p* < 0.05, than students who completed the study.

#### 2.2.1. Intervention Group

Students at University Burgundy Franche-Comté were informed by email from the university health and social services of the possibility of taking part in an online program that was broadly described as helping students learn life skills to enhance their university experience. Participants who were interested in the program were then invited to answer an online survey after signing a consent form. Once they had completed the survey, participants were given access to an e-learning module of the ETUCARE program every week during the 8-week intervention period and received an email explaining the content of the upcoming module. After the 8-week period, participants were sent a link to complete the online post-intervention questionnaire (T1).

#### 2.2.2. Control Group

Meanwhile, an associate professor from University Burgundy Franche-Comté sent an email to all psychology students at the university inviting them to participate in an online longitudinal study exploring students’ psychological state during the COVID-19 pandemic. Interested students were asked to answer an online survey after signing a consent form. After an 8-week period, they received an email asking them to complete an online survey again. At the end of the post-intervention survey, we screened them to ensure for the possibility that they had not taken part in the ETUCARE program. Participants who answered “yes” were excluded from data analysis. Indeed, given the major mental issues observed during the pandemic in university students, we decided not to randomize our sample by placing participants on a waiting list and considered that it was important that any student who wished to take part to the program could do so. The different stages of the study are illustrated in Figure 1.

### 2.3. Intervention

ETUCARE is an 8-week, web-based mental health program, which was specifically designed for university students during the COVID-19 pandemic. The ETUCARE program was co-designed with university students using a participatory approach. First, an online survey and student interviews were conducted to identify students’ mental health issues and needs and define the program features (unpublished results). Then, three master’s students (public health, psychology, and management-marketing) were involved in the creation process, pre-tested the modules, and helped improve their design so that they would be attractive to other students. The different modules were designed by a full professor; an expert in clinical psychology, mindfulness-based programs, and positive psychology interventions; a PhD in psychology and a psychologist; an expert in cognitive and behavioral therapy and emotion regulation in young people; and three project managers in public health, who are experts of preventive and participatory approaches. The program was the result of this co-design effort between the experts and the students The ETUCARE program integrated evidence-based strategies from CBT, positive psychology, mindfulness, and lifestyle medicine through eight e-learning modules featuring the following themes: mental health information, stress management, procrastination and motivation for learning, sleep and insomnia, self-awareness, emotion regulation, meaningful relationships, and booster session (see Table 1 for a detailed description of the ETUCARE program). The selection of the themes of the modules was based on the results of the student’s interviews that were conducted during the participatory approach and on a systematic review on like and dislike contents in digital mental health programs [64].

Each e-learning module had a duration of 45 min and contained psychoeducation information, videos, student experiences, tools, quizzes (e.g., sleeping habits, identifying stressors, etc.), and a number of weekly exercises based on evidence-based practices. Exercises took 5 to 15 min, enabling them to be carried out within busy academic schedules. Each module approached elements and examples specific to the COVID-19 pandemic (e.g., themes of worry, stressors associated to the pandemic, distance learning courses, social restrictions). No face-to-face contact occurred during the study, but e-mails were sent to notify and remind participants of the upcoming modules. The latest version of the program is available via the website https://elearning.ireps-bfc.org/ (accessed on 15 May 2022).

### 2.4. Measures

The first section was composed of validated self-report questionnaires measuring mental health problems (depression, anxiety, psychological distress, insomnia, and alcohol consumption) and psychological well-being. The second part collected the students’ socio-demographic information. All measures were completed online before (T0) and immediately after (T1) the intervention.

**Demographic information:** Participants provided information about their gender, age, place of residence, and field of study.

#### 2.4.1. Kessler Psychological Distress Scale (K6)

The Kessler 6-item is a shortened version of the K-10 scale, which is a global measure of psychological distress based on depression and anxiety symptoms. The maximum score is 24. A score of 13 or higher indicates the likelihood of a psychological disorder, while a score of less than 13 indicates that severe mental illness is unlikely; studies generally use a score of 8 to 12 to indicate moderate distress [65]. In the present study, this scale showed good internal consistency (Cronbach’s alpha T0 = 0.84; T1 = 0.87).

#### 2.4.2. The Generalized Anxiety Disorder 2-Item (GAD-2)

The GAD-2 is an ultra-brief version of the seven-item GAD-7 scale, with two questions relating to critical components of any anxiety disorder. Scores on the GAD-2 scale range from 0 to 6. A score of 3 or more is the preferred threshold for identifying clinically significant symptoms of generalized anxiety disorder. Using a cut-off score of 3, the GAD-2 has a sensitivity of 86% and a specificity of 83% for the diagnosis of generalized anxiety disorder [66]. The psychometric properties in this study were satisfactory (Cronbach’s alpha T0 = 0.78; T1 = 0.84).

#### 2.4.3. Patient Health Questionnaire 2-Item (PHQ-2)

The PHQ-2 assesses the frequency of depressed mood and anhedonia over the past two weeks. It comprises the first two items of the PHQ-9. Scores on this scale range from 0 to 6. A score of 3 or more is the preferred threshold for identifying clinically significant symptoms of major depression. Using a cut-off score of 3, the PHQ-2 has a sensitivity of 83% and a specificity of 92% for the diagnosis of major depression [67]. The psychometric properties in this study were satisfactory (Cronbach’s alpha T0 = 0.67; T1 = 0.73).

#### 2.4.4. The Insomnia Severity Index (ISI)

The ISI is a seven-item self-report questionnaire assessing the nature, severity, and impact of insomnia [68]. The usual recall period is “the past month”, and the dimensions evaluated are: severity of sleep onset, sleep maintenance, early morning awakening problems, sleep dissatisfaction, interference of sleep difficulties with daytime functioning, noticeability of sleep problems by others, and distress caused by the sleep difficulties. A 5-point Likert scale is used to rate each item (e.g., 0 = no problem; 4 = very severe problem), yielding a total score ranging from 0 to 28. The total score is interpreted as follows: absence of insomnia (0–7); sub-threshold insomnia (8–14); moderate insomnia (15–21); and severe insomnia (22–28). The psychometric properties in this study were satisfactory, except at T1 (Cronbach’s alpha T0 = 0.76, T1 = 0.57).

#### 2.4.5. The Alcohol Use Disorders Identification Test-Concise (AUDIT-C)

The Alcohol Use Disorders Identification Test-Concise (AUDIT-C) is a brief screening instrument that reliably identifies individuals who are hazardous drinkers or have active alcohol use disorders (including alcohol abuse or dependence). The AUDIT-C is a modified version of the 10-question AUDIT instrument; it has 3 questions and is scored on a scale of 0–12. Each question has 5 answer choices valued from 0 to 4 points. In men, a score of 4 or more is considered positive, which is optimal for identifying hazardous drinking or active alcohol use disorders. In women, a score of 3 or more is considered positive. Generally, the higher the score, the more likely it is that a person’s drinking habits affect his or her safety [69]. The psychometric properties in this study were satisfactory (Cronbach’s alpha T0 =  0.83; T1 = 0.77).

#### 2.4.6. The Warwick-Edinburgh Mental Well-Being Scale (WEMWBS)

The WEMWBS [70] is composed of 14 positively-worded items (e.g., “I’ve been feeling optimistic about the future”). Each item is rated on a 5-point Likert scale, ranging from 1 (none of the time) to 5 (all of the time). The higher the mean score, the greater the level of well-being. In the present study, this scale showed good internal consistency (Cronbach’s alpha T0 = 0.88; T1 = 0.88).

## 3. Results

### 3.1. Mental Health Problems at T0

Table 2 presents the frequency of psychological problems observed in the student population of Bourgogne Franche-Comté in February 2021.

In February 2021, one-third (34%) of the students in our sample met clinical levels of severe psychological distress, and 58% presented symptoms of moderate psychological distress. In other words, only 8% of students did not show any symptoms of psychological distress. These rates are in line with the main national surveys of French student populations [24,25] carried out between April and December 2020, in which rates of psychological distress ranged from 22% to 60% depending on the scales and thresholds used. In our sample, 54% of students met clinical levels of generalized anxiety symptoms and 43% reported clinical levels of major depression; in the national surveys [24,25,26,71], the proportion of students reporting anxiety and depression ranged from 24–38% and 16–43% respectively. Sleep problems were reported by 71% of students in our sample, with 34% showing clinical insomnia; these rates are similar to those observed in the COVIPREV survey for the same period (73%) [71]. Regarding alcohol use, 39% of our students reported hazardous use, which is comparable to the 40% of medical students identified in the BOURBON survey [5].

### 3.2. Efficacy of the Web-Based Program on Student Mental Health Problems and Well-Being

To test our hypothesis that the ETUCARE program would be effective in reducing student’s mental health problems (psychological distress, depression, anxiety), sleep problems, and alcohol use, as well as increasing psychological well-being, ANOVAS were conducted on scores with Time (T0, T1) as the within-participant factor and Intervention (ETUCARE, Control) as the between-participant factor. A significant interaction between Intervention and Time indicates that there was a differential effect between the intervention and the control condition on an outcome.

Analyses were made on complete cases and missing data were handled using pairwise deletion. Descriptive statistics (means and standard deviations) for all measures at T0 and T1 for the ETUCARE program and the control group and 2 × 2 mixed ANOVAs are presented in Table 3.

Descriptive statistics showed that participants in the intervention group presented higher scores of psychological distress and depression and lower scores of well-being, alcohol use, and sleep problems than participants in the control group at baseline.

The 2 × 2 mixed ANOVAs revealed a significant Time x Intervention interaction for well-being scores, *F* (1, 56) = 4.45, *p* = 0.04. The results indicate that, compared to the control group, participants assigned to the ETUCARE group showed a significant increase in well-being between T0 (*M* = 36.3, *SD* = 7.67) and T1 (*M* = 40.10, *SD* = 9.47). The results also revealed a marginally significant Time x Intervention interaction for depression scores, *F* (1, 56) = 3.41, *p* = 0.07. Compared to the control group, participants in the ETUCARE group showed a decrease in depressive symptoms between T0 (*M* = 2.90, *SD* =1.68) and T1 (*M* = 2.20, *SD* = 1.58). The results did not reveal any differential effect between the intervention and the control condition on other outcomes.

### 3.3. Clinical Significance

We also report results for the subsamples of participants presenting clinical significance at T0, i.e., participants with scores above cut-off levels for clinical measures of psychological distress, depression, anxiety, insomnia, and hazardous drinking. Chi-squared tests were used to compare the frequency of clinical significance between T0 and T1 in each group (Etucare vs. Control). The results revealed a significant reduction between T0 and T1 of clinical cases of severe psychological distress in the ETUCARE group compared to the control group, *Χ*^2^ (1) = 3.82, *p* = 0.05. Among participants with clinical levels of severe psychological distress at T0, 42.9% showed clinical improvement in their symptoms after completing the ETUCARE program (i.e., score below cut-off levels at T1), versus 0% in the control group. The results also revealed a significant reduction between T0 and T1 of clinical cases of anxiety in the ETUCARE group compared to the control group, *Χ*^2^ (1) = 3.67, *p* = 0.05. Among participants with clinical levels of generalized anxiety disorder at T0, 20% showed clinical improvement in their symptoms after completing the ETUCARE program, versus 0% in the control group. Finally, the results revealed a significant reduction between T0 and T1 of clinical cases of hazardous drinking in the ETUCARE group compared to the control group, *Χ*^2^ (1) = 5.30, *p* = 0.02. Among participants with hazardous drinking levels at T0, 28.6% showed clinical improvement of their symptoms after completing the ETUCARE program compared to 0% in the control group.

There was no significant reduction in clinical caseness between T0 and T1 for other outcomes. Among participants with clinical levels of major depression at T0, 36.4% showed clinical improvement of their symptoms after completing the ETUCARE program compared to 33.3% in the control group. Among participants with symptoms of sleep problems at T0, 33.3% showed clinical improvement after completing the ETUCARE program compared to 11.5% in the control group.

## 4. Discussion

The aim of this study was to test the effectiveness of an online multicomponent self-help program to promote university students’ mental health during the COVID-19 pandemic. We hypothesized that mental health problems (i.e., psychological distress, depression, and anxiety), sleep problems, and alcohol use would be significantly reduced, and well-being would be significantly increased in students who participated in the web-based ETUCARE program compared to the control group.

At T0, the prevalence rates of severe distress, depression, anxiety, clinical insomnia, and hazardous drinking were 34%, 54%, 43%, 34%, and 39%, respectively. The rates for psychological distress, sleep disorders, and alcohol use are comparable to those observed in French national surveys conducted during the pandemic crisis (CN2R, CONFINS, COVER, COVIPREV), but the prevalence of depressive and anxiety disorders was higher in our sample. This difference could be explained by our use of ultra-brief versions of the GAD-7 and PHQ-9, whose diagnostic threshold is based on the two items that best represent the disorder, which may increase sensitivity as well as false-positive rates [72].

Repeated-measures ANOVAs revealed a significant increase in students’ well-being and a marginally significant decrease in students’ depressive symptoms after the 8-week ETUCARE intervention compared to the control group. These results are in line with recent research that showed that online interventions can lead to both a reduction in mental health problems and an increase in well-being [55] in university students in the context of the COVID-19 pandemic. As previous studies [73] observed a greater intervention effect for participants with problematic mental health scores at baseline, we were interested in investigating the intervention effect on clinical significance. The results revealed a significant reduction between T0 and T1 of clinical caseness of psychological distress, anxiety, and hazardous drinking in the intervention group compared to the control group. Thus, these findings suggest the beneficial effect of the ETUCARE program for improving university students ‘mental health at both clinical and non-clinical levels. However, the current results should not be applied to a general population cohort; extrapolation to clinical populations should be used with caution. Indeed, although the sample did include participants that showed high distress, depression, and anxiety levels, the presence of psychological symptoms does not equal a diagnostic disorder. Contrary to our hypothesis, our program had no effect on students’ sleep problems. We can assume that the program, with only a single module on sleep, was insufficient to help students to mobilize new resources (e.g., stimulus control, sleep hygiene) to manage their sleep. Indeed, the duration of online intervention for sleep improvement is typically 5 to 9 weeks [74].

To our knowledge, our study is the first to have assessed the effects of an innovative multicomponent program that integrated various evidence-based strategies and targeted students’ mental health during the COVID-19 pandemic. Our findings suggest that this online program may constitute a promising pathway to buffer the mental health consequences of the COVID-19 pandemic in such a vulnerable population. Another strength of the study is that the intervention used an e-learning management system and was delivered without reliance on a clinical staff. Indeed, mental health systems are typically under-resourced, and were further deteriorated during COVID-19. Positive results for interventions that promote mental health in both outcomes of well-being and mental health problems makes it possible to consider such an approach as a promising standalone first-line intervention or as a solution to deal with existing system issues (e.g., waitlists). Albeit promising, the results of the present study should be treated with caution and further research efforts will be needed to confirm the beneficial effects of this type of program on university students. In addition, although our results indicated a short-term intervention effect on student’s mental health and well-being, further research efforts will be needed to assess its long-term impact using follow-up assessments after the intervention.

The limitations of this study included sample characteristics. Our sample was small and was predominantly female, which limited the generalization of results. In addition, recruitment was carried out on an exclusively voluntary basis, and hence participants may have had greater motivation and higher readiness for change than the general student population. Furthermore, the intervention was delivered to any student who signed on, rather than randomizing them into waitlists in order to propose a response to the immediate mental health demands in the student community during the pandemic. Thus, it is important to compare and verify the reliability between an active intervention and a comparable control condition in a randomized controlled trial in future studies.

A large proportion of students in the intervention group dropped out (66%) from this study, and this limits the generalizability of the results. Indeed, bias due to missing outcome data may lead to a poor estimation of the intervention effect [75]. Although the high attrition rate is consistent with the literature on web-based intervention trials [76,77], we expected fewer students would drop out because we designed the ETUCARE program using a participatory approach. Indeed, it has been shown that digital mental health interventions that are tailored to meet the needs of specific populations can improve relevance, engagement, and efficacy [64,78]. Despite this participatory approach, it is still possible that the length and format of the intervention did not appeal to all students and contributed to attrition. Strategies that increase participant engagement, such as gamification strategies and rewards through the awarding of points for the successful completion of challenges [79], are imperative to optimize the program’s outcomes and effectiveness. Further research is required to elucidate the factors that may or may not contribute to participants’ engagement in an online mental health intervention and to better inform the design of such interventions to optimise engagement.

It is also a limitation of this study that measures of engagement with e-learning modules and evidence-based strategies were not captured. This limitation prevented us from investigating the relative contribution of the various evidenced-based strategies to the positive mental health outcomes. Further research is required to investigate the relative contribution of the various evidenced-based strategies incorporated into the intervention and their compounding effect and to understand the underlying mechanisms of the effect observed in the present study. With this aim, future studies could use smartphone applications to conduct ecological momentary assessments and measure how mental health risks and protective factors evolve after using various evidence-based strategies during and after the intervention [80]. Furthermore, evidence suggests that the effects of web-based mental health programs are larger in subgroups of participants that have symptoms or risk factors [48]. Thus, more research is needed to identify specific subsets of students who respond best to the intervention, which types of evidence-based strategies are most suitable for which students, and to explore ways to optimize intervention effects and thus fully exploit their potential to improve university students’ mental health. Future studies should address these limitations.

## 5. Conclusions

The COVID-19 pandemic and the abrupt closures of campuses led to the loss of daily routines and had a major impact on academic studying, exacerbating academic stressors and mental health problems in university students. It seems crucial for universities to take steps to prevent any further mental health deterioration and promote well-being in university students. Our pilot study highlighted the promising effects of an online self-help program for promoting well-being and preventing mental health problems in university students. More specifically, the results revealed an increase in well-being and a reduction of clinical cases of psychological distress, anxiety, and hazardous drinking among students in the intervention group. The results herein bolster the evidence that online interventions can play a significant role in dealing with the mental health burden of the pandemic [46,47]. Further research efforts will be needed to confirm the beneficial effects of this type of program on university students, and to explore underlying mechanisms and the duration of intervention effect over time.

## Figures and Tables

**Figure 1 ijerph-19-10442-f001:**
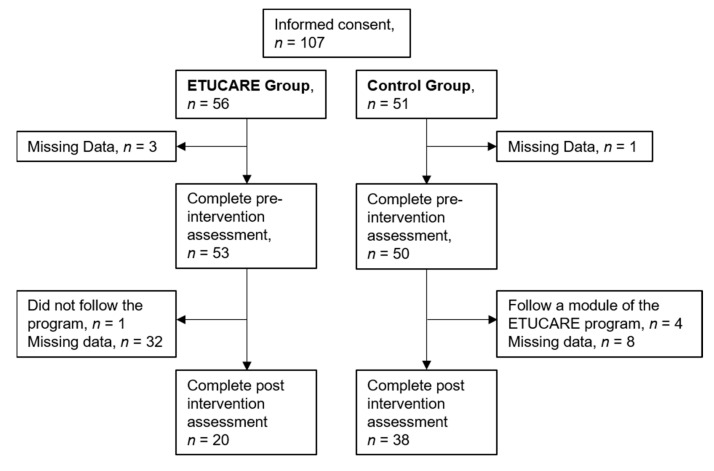
Study flow chart.

**Table 1 ijerph-19-10442-t001:** Program description.

Theme	Content	Weekly Panel of Exercises and Activities
Module 1: Mental Health Information	Definition of mental health, positive mental health, and common psychological problems Mental health risk and protective factors How and where to seek help for a mental health problem How to take care of one’s own mental health Program schedule	Online mental health self-assessment quizzes
Module 2: Stress management	Definition and explanation of stress and stressors Impact of stress on our body and feelings. Effect of physical activity and nature (e.g., birdsong, forest) on stress Relaxation strategies to cope with stress	Body scan meditation Breathing control exercise Progressive muscle relaxation Increasing “recharge” activities Increasing physical activity Increasing immersion in natural environment Problem resolution
Module 3: Procrastinaton and Motivation for Learning	Explanation of procrastination in everyday life Basic procrastination triggers Self-assessment tools to identify one’s own procrastination behaviors Definition of self-blame and its effects Misconceptions about failure and perfectionism Processes involved in motivation	Self-compassion exerciseMindful breathingSetting smart goals
Module 4: Sleep and insomnia	Definition and explanation of sleep and insomnia Introduction to the concept of the link between thoughts, feelings, and behaviour. Link between sleep, physical activity, and diet Self-assessment tools on sleep hygiene and sleep quality	Mindfulness meditation Worry journal Sleep hygiene Stimulus control for insomnia
Module 5: Self-Awareness	Introduction to the concept of self-esteem and to activities that could help to enhance self-esteem. Identifying and using one’s strength Identifying and reconnecting to one’s value	VIA Character Strengths Survey Cultivating one’s strengths Vision board Engagement in meaningful actions
Module 6: Emotion regulation	Definition and explanation of emotions and their usefulness in daily life Explanation of the link between thoughts, feelings, and behaviour Introduction to thinking errors and how to manage them Mood tracker tool Emotion regulation self-assessment Link between healthy behaviors and mood Introduction to positive emotion and strategies to cultivate positive emotions in daily life	Cognitive reappraisal Tolerating distress Reorientation of attention towards the positive and satisfying aspects of life Gratitude wall Savouring the present moment
Module 7: Meaningful Relationships	Discuss the importance of having a good relationship. Learn specific skills that are needed to get along with other people (communication skills and assertiveness) Assertiveness self-assessment Learn to deal with social problem	Random acts of kindness Reorientation of attention towards the positive in relationships Active listening Assertiveness exercise Gratitude letter
Module 8: Booster	Presentation of the highlights of the different modules	Continue exercises found to be helpful.

**Table 2 ijerph-19-10442-t002:** Prevalence of psychological problems observed in the student population of Bourgogne Franche-Comté in February 2021 compared to French national survey data.

	ETUCARE Group (*n* = 53)	Control Group (*n* = 50)	Total Sample (*N* = 103)	Main National Surveys
**Severe psychological distress**	40%	27%	34%	COVER: 60% CN2R: 22%
**Anxiety**	57%	51%	54%	COVER: 38% CONFINS: 24% CN2R: 28% COVIPREV: 34%
**Depression**	53%	33%	43%	COVER: 23% CONFINS: 33% CN2R: 16% COVIPREV: 43%
**Sleep problems**	74%	68%	71%	COVIPREV: 73%
**Hazardous drinking**	32%	47%	39%	BOURBON: 40%

Periods of data collection of the main national surveys. COVER survey: December 2020; CN2R survey: April 2020; CONFINS survey: May 2020; COVIPREV survey: February 2021; BOURBON survey: May 2017.

**Table 3 ijerph-19-10442-t003:** Descriptive statistics (means and standard deviations) for all measures at T0 and T1 for the ETUCARE program and the control group and 2 × 2 mixed ANOVAs.

	ETUCARE Group		Control Group		Control Group vs. ETUCARE Group	
	T0	T1	T0	T1		
	M (SD)	M (SD)	M (SD)	M (SD)	*F* (1, 56)	*p*
Kessler 6	11.00 (4.34)	10.50 (6.64)	9.11 (4.33)	9.97 (4.97)	2.16	0.14
Gad 2	2.90 (1.83)	3.00 (1.86)	2.79 (1.97)	3.34 (1.92)	1.46	0.23
PHQ 2	2.90 (1.68)	2.20 (1.58)	1.84 (1.46)	1.79 (1.58)	3.41	0.07
WEMWBS	36.30 (7.67)	40.10 (9.47)	41.50 (8.76)	41.76 (7.87)	4.45	0.04
ISI	9.30 (5.78)	9.90 (3.78)	11.26 (5.00)	10.24 (3.96)	1.89	0.17
Audit	1.75 (1.89)	1.55 (1.93)	2.92 (2.77)	3.13 (2.62)	1.52	0.22

## Data Availability

Data is available upon request to the corresponding author.

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
