# Peer review of "Promoting University Students’ Mental Health through an Online Multicomponent Intervention during the COVID-19 Pandemic"

_ijerph, 2022, doi:10.3390/ijerph191610442_

Round 1

Reviewer 1 Report

Review of the manuscript:

Type of manuscript: Article
Title: Promoting university students’ mental health through an online
multicomponent intervention during the COVID-19 pandemic
Journal: International Journal of Environmental Research and Public Health

 Introduction

1.The introduction addresses high levels of academic burnout, but is missing background information for this issue, namely among university students. (p.68)

2. The paper addresses a number of studies thar investigated different types of online mental health programs for university students, but there is no information about how these programs are structured and how they can help prevent depression, anxiety, and alcohol-related problems (p.93)

3. I did not find information about what means multitheoretical intervention (p.109)

4. Please specify the participation of university students in the ETUCARE program when you said that the program was co-designed with university students using a participatory approach. (p.122)

2. Methods

1. There is no theoretical evidence of the importance of including these domains in the ETUCARE program positive psychology, mindfulness and lifestyle medicine through eight e-learning modules featuring the following themes: mental health information, stress management, procrastination and motivation for learning, sleep and insomnia, self-awareness, emotion regulation, meaningful relationships and booster session (p. 185)

2. Who participated in the study? What were the study inclusion and exclusion criteria? Do you mean of approximately the same age group? What gender difference were found?

 3. Results/Discusion

1. We hypothesized that students who participated in the web-based ETUCARE program would show greater decreases in mental health problems, sleep problems and alcohol use. But, the participants were those students with large problems. Explain better.

Please explain better the results/discusion found in Kessler 6, Gad 2, WEMWBS, with ETUCARE Group and Control Group. The explanation of the results is confused. (p.32)

Author Response

We appreciate the time and effort that you dedicated to providing feedback on our manuscript and are grateful for the insightful comments on and valuable improvements to our paper.
We have incorporated the suggestions you have made. Those changes are highlighted within the manuscript. Please see below, in blue, for a point-by-point response to your comments. All line numbers refer to the revised manuscript file with tracked changes

Reviewer 2 Report

Comments and suggestions for Authors

The manuscript entitled ‘Promoting university students’ mental health through an online multicomponent intervention during the COVID-19 pandemic’ refers to the assessment of an online intervention program to improve the students’ mental health in the context of the Covid-19 pandemic.

I appreciate the work of the authors and the implementation of a digital program based on various strategies (psychological, lifestyle medicine) and addressed to students to help them to buffer the negative effects of the pandemic and to keep a good mental health.

The manuscript is interesting and is presented in an organized, systematized manner.

 Introduction

It is well written, but the authors should cite more recent references.   

 Methods

The Figure 1 (Study flow chart) is illustrative, but, when referring to the control group, it would be preferable to specify T1, not “post intervention” (according to the exclusion criteria, participants from the control group had not taken part in the program and did not benefit from any intervention). This is also valid in other parts of the manuscript (e.g. Table 3, lines 305-306, 314, etc.).  Once it is stated that the evaluation of the participants was carried out before and after the interval corresponding to the online intervention, maybe the authors should refer to T0 and T1.

The ETUCARE program description is clearly presented, but the authors should mention who developed this program (e.g. professionals in different fields). In the manuscript is written that it was co-designed with university students using a participatory approach’ and ‘3 master’s students’ were involved in this process.

Results

The authors should present in more detail how they proceeded with the statistical analysis, considering the dropout rate especially in the intervention (ETUCARE) group (62%). The authors should specify which method was used to analyze the missing data.

Discussion

The limitations of the study are presented realistically, as well as the innovative aspects of a multicomponent interventional online program targeting the mental health of students.

 References

Out of a total of 69 references, 25 are prior to 2017. The authors should improve the references, especially those related to the Introduction, without being limited to them (e.g., regarding dropout, there are more current references than reference no. 65).

Review date: 08.08.2022

Author Response

(The authors gave the same response as above.)
